# Inverted Basins by Africa–Eurasia Convergence at the Southern Back-Arc Tyrrhenian Basin

**Maria Filomena Loreto [1,\*], Camilla Palmiotto [1], Filippo Muccini [2,3] , Valentina Ferrante [4] and Nevio Zitellini [1]**

1   National Research Council, CNR, Marine Sciences Institute, ISMAR, 40129 Bologna, Italy;
    Camilla.palmiotto@bo.ismar.cnr.it (C.P.); nevio.zitellini@bo.ismar.cnr.it (N.Z.)
2   Istituto Nazionale di Geofisica e Vulcanologia, INGV, 00143 Roma, Italy; filippo.muccini@ingv.it
3   National Research Council, CNR, Istituto di Geologia Ambientale e Geoingegneria, IGAG, 00010 Roma, Italy
4   Independent Researcher, 40131 Bologna, Italy; valentina.ferrante@bo.ismar.cnr.it
\*   Correspondence: filomena.loreto@bo.ismar.cnr.it; Tel.: +39 0516398878

**Abstract:** The southern part of Tyrrhenian back-arc basin (NW Sicily), formed due to the rifting and spreading processes in back-arc setting, is currently undergoing contractional tectonics. The analysis of seismic reflection profiles integrated with bathymetry, magnetic data and seismicity allowed us to map a widespread contractional tectonics structures, such as positive flower structures, anticlines and inverted normal faults, which deform the sedimentary sequence of the intra-slope basins. Two main tectonic phases have been recognised: (i) a Pliocene extensional phase, active during the opening of the Vavilov Basin, which was responsible for the formation of elongated basins bounded by faulted continental blocks and controlled by the tear of subducting lithosphere; (ii) a contractional phase related to the Africa-Eurasia convergence coeval with the opening of the Marsili Basin during the Quaternary time. The lithospheric tear occurred along the Drepano paleo-STEP (Subduction-Transform-Edge-Propagator) fault, where the upwelling of mantle, intruding the continental crust, formed a ridge. Since Pliocene, most of the contractional deformation has been focused along this ridge, becoming a good candidate for a future subduction initiation zone.

**Keywords:** inverted basins; contractional tectonics; extensional tectonics; STEP fault; back-arc region; Tyrrhenian basin

## 1. Introduction

Back-arc basins (BABs) are a realm of extensional tectonics controlled by the subduction and roll-back of oceanic lithosphere [1], where normal faults, listric or planar [2], dominate. Lithosphere extension may lead to the thinning of continental margins up to the crustal breakup and to the mantle upwelling and/or formation of new oceanic crust [3,4]. Usually, this process ends when continental crust carried by the downgoing plate enters into the subduction zone [5], the slab retreating slows down or stops, and the back-arc basins may start to be inverted, as inferred by studying the Pannonian Basin [6] or the Carpathians belt [7]. Contractional deformation has been widely observed within back-arc basins and is usually associated with an inversion phase (eastern Japan Sea; [8]) or with an oblique subduction system able to trigger polyphase deformations involving transtension and transpression (Salin Subbasin; [9]). Tectonics of BABs can be further complicated by the presence of large strike-slip faults, like scissor rupturing the crust, generated by the lateral termination of the subducting slab [10]. These faults, called STEP (Subduction-Transform-Edge-Propagator) faults, are observed within BABs with irregular shape and are laterally confined by large strike-slip systems [11], for instance, as in the North Fiji Basin, where a long strike-slip fault bounds the system to the south [12], or in the Eastern Caribbean plate, bounded to the south by a strike-slip fault currently considered active [13].

The Tyrrhenian basin (Figure 1A) embodies all the tectonic elements described above. It is a small back-arc basin, controlled by the subduction and roll-back of Ionian oceanic

lithosphere [14–16], that opens in response to the eastward migration of the overriding Calabrian Arc occurring while Africa and Eurasia plates are converging [17,18]. Opening of the Tyrrhenian BAB started with the formation of a series of listric and planar conjugate faults, west of Sardinia, during Middle Miocene, which was followed by continental breakup and crustal accretion/mantle exhumation [19–22] during Upper Miocene and Lower Pliocene. The Vavilov Basin started to open during Upper Messinian/Lower Pliocene time and stopped in the upper Pliocene; afterward, extension moved to southeast and induced the opening of the Marsili Basin, started during the lower Pleistocene (2 Ma) [23–25]. The main extensional character of the Tyrrhenian BAB is well documented by the recent morpho-tectonic compilation of [25], even if a recent re-organization of the stress field within the Tyrrhenian Basin has been hypothesised [26,27] to account for the contractional tectonic structures detected along the eastern and southern Tyrrhenian Margins.

Others works have highlighted that the Tyrrhenian BAB is a more complex system, within which extensions co-exist or are overprinted by contractional and strike-slip systems [28–30]. Sporadic contractional events, represented by anticline at the top of inverted normal faults, have been already recognized within some basins at the rear of the arc [28,31,32]. These features, observed along the southern Tyrrhenian margin, are hypothesized to be associated with the beginning of the back-arc basin inversion [27]. This was also supported by [33], a study that, based on morpho-bathymetric data analysis acquired in the Marsili Basin, concluded that the extension in this Basin has not been active since about 1 Ma.

Nevertheless, subduction may be currently active as suggested by the recent discovery of ca. 3000-year-old Tephra recovered from the top of the volcano Marsili [34] and by the magmatic activity in the Aeolian Volcanic Arc [35,36].

Transcurrent E–W trending faults bounding southward the Tyrrhenian Basin have been initially proposed by [37]. Ref. [36] suggested that slab tearing controls the super inflate Marsili volcanic ridge and defined the location of a paleo-STEP fault bounding southward the Aeolian Volcanic Arc. Later, [38] proposed a reconstruction of the evolution of the subduction system in the Tyrrhenian, suggesting the presence of two main STEP faults, bounding northward and southward the subducting slab under Calabria. The STEP fault bounding the southern edge of the subducting slab is hypothesized to pass northward of Sicily [11,37,38], entering in the Ionian Sea [39,40]; while [41] hypothesizes the STEP fault bounding the northern edge of the subducting slab corresponds to the E–W trending Palinuro Volcanic Complex.

Southern Tyrrhenian, together with Apennine Chain, is the locus of intense seismicity (Figure 1B). The seismicity recorded under the Calabria region is generated by normal fault systems accommodating the up-lift of the continental block [42–44], while the seismicity recorded in its western offshore is generated by the subducting slab as inferred by the depth of events [45]. Seismicity to north of Sicily strongly decreases, if compared to Calabria, and the focal mechanisms of major recorded events have been associated with compression [46]. In this part of the continental margin, seismicity associated with extension has been also recorded, suggesting that there is an area in which compression co-exists with extension [47,48].

In this paper we define the tectonic processes associated with the beginning of the Tyrrhenian back-arc inversion by analyzing the compressive structures detected in the northern sector of the Sicilian continental margin. Basin inversions due to contractional phases following extensional phases have been already observed around the world [7–9,49] and in some cases have been controlled by a rapid reconfiguration of plate boundaries [50]. We focused on the southern part of the Tyrrhenian back-arc basin by merging all available geophysical data—-single and multi-channel seismic profiles as well as multibeam and magnetic data—-with information available in literature on seismicity, numerical modelling, and geodetic information in order to define the active tectonic structures as well its evolutionary model for the Plio-Quaternary, which may help with hazard assessment of the emerged areas.

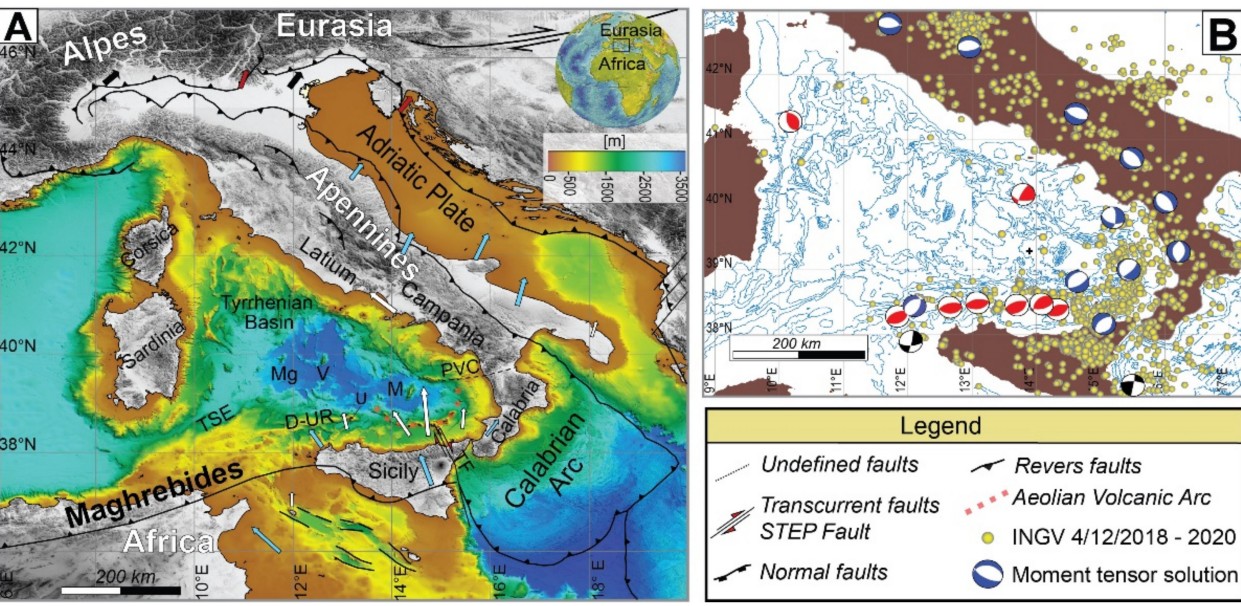

**Figure 1.** (**A**) Geodynamic map of the Apennine subduction system. The shaded relief of bathymetry is produced by gridding data freely downloaded from EMODnet portal (EMODnet Bathymetry Consortium, 2018), while the Digital Elevation Model (DEM) is produced gridding data from SRTM90 (http://srtm.csi.cgiar.org, accessed on 31 May 2016). Structural features are synthesized in accordance with [51]. Thick arrows are: light blue, the displacement vectors between Africa with respect Eurasia from GPS measurements [52]; red, the GPS measurements from the free accessible website (https://www.unavco.org, accessed on 11 August 2016); white and black, GPS residual velocity [53]. (**B**) Map of seismicity: earthquake epicentres (small yellow dots) recorded from 12/2018 to 12/2020, downloaded from European Mediterranean Seismological Centre (EMSC) portal (https://www.emsc-csem.org/Earthquake/, accessed on 23 April 2020); moment tensor solution included in the Italian Centroid-Moment-Tensor (CMT) database and modified from [54–56]. Red balls indicate compression, blue balls indicate extension and black balls indicate strike-slip. D-UR: Drepano-Ustica Ridge; Mg: Magnaghi; V: Vavilov; M: Marsili; U: Ustica; TSE: Tunisia-Sardinia Escarpment; TF: Tindari Fault; PVC: Palinuro Volcanic Complex.

## 2. Materials and Methods

### 2.1. Morpho-Bathymetry

Middle resolution bathymetric data (115 × 115 m-cell grid size) were downloaded from EMODnet (European Marine Observation and Data Network) bathymetry portal (https://www.emodnet-bathymetry.eu/, accessed on 7 December 2020). The EMODnet harmonized data set resulted from the merging of modern high-resolution bathymetry collected around the Tyrrhenian basin (MAGIC Project supported by the Italian Civil Protection Dept.) with older multibeam data [57], while further data coverage gaps were bridged by integrating the General Bathymetric Chart of the Oceans (GEBCO; https://www.gebco.net/data_and_products/gridded_bathymetry_data/gebco_30_second_grid/, accessed on 7 December 2020). We merged and gridded, with the "nearest neighbour" algorithm, data using the open source software GMT (Global Mapper Tool; [58]). The datum and projection used are WGS84 and coordinate longitude and latitude, respectively. The resulting morpho-bathymetric grid was exported as a high-resolution GeoTIFF raster image and then merged with seismic profiles in a Kingdom (HIS Markit) project (Figure 2A).

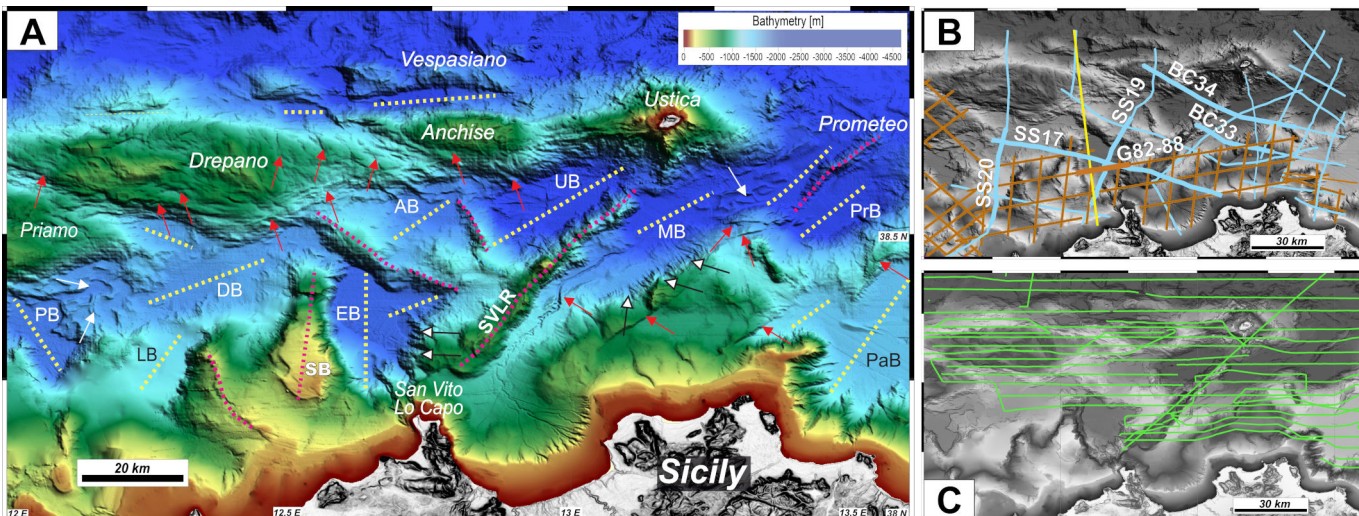

**Figure 2.** (**A**) Morpho-bathymetric map of NW Sicilian margin, showing the main morphological features. PB: Priamo Basin; LB: Leda Basin; DB: Drepano Basin; AB: Anchise Basin; UB: Ustica Basin; MB: Medea Basin; PrB: Prometeo Basin; PaB: Palermo Basin; SVLR: San Vito Lo Capo Ridge; EB: Erice Basin; SB: Scuso Basin. Red arrows point to morphological escarpments; white and black arrows point to slope scars; white arrows point to small undulated ridges. (**B**) Location map of seismic profiles. Light blue lines: Sparker profiles; yellow line: Crop profile; brown lines: seismic reconnaissance. Thick lines indicate that profiles are shown in figures below. (**C**) Location map of magnetometric profiles.

### 2.2. Seismic Profiles

The seismic profiles used in this work belong to different datasets. The high-resolution seismic sections were collected on board R/V Bannock in the seventies and eighties [59] as a part of a multidisciplinary project ("Progetto Finalizzato - Oceanografia e Fondali Marini") supported by the Italian National Research Council (CNR). All the seismic lines were designed and carried out with directions both parallel and perpendicular to the continental margin to better investigate the deep structures and reveal the complex geodynamic evolution of the area. A Sparker 30 kJ seismic source from Teledyne Exploration of Houston (USA) was used, with a shooting interval of 15 minutes (about 25 m), while the high-resolution single channel profiles were recorded down to 4–8 s TWT (Two Way Time) depth bsl (below sea level) with an analog system and printed in real time on paper. The navigation system used at the time was the Loran C, a hyperbolic radio navigation system that allowed a receiver to determine its position by listening to low frequency radio signals transmitted by fixed land-based radio beacons. Radio signals were then converted into the ED50 coordinates system through a non-automatic procedure. For the purposes of this work, a selection of seismic lines (light blue lines in Figure 2B) was scanned from paper to high-resolution raster image (gray scale 8 bit 300 dpi BMP). To convert a raster image into a georeferenced SEGY (a standard-format developed to exchange geophysical data) seismic profile, both navigation files and the seismic images were processed. The positioning errors due to the "vintage" Loran C navigation system were reduced to a minimum through an interactive procedure. The data were then converted to the UTM33-WGS84 reference system. The seismic images were interactively digitized to define the x and y pixel coordinates used in the conversion process. For better georeferencing, the Shot Point displayed in the bitmap sections were digitized, and the corresponding pixel values were associated with the real coordinates in the navigation files. The seismic images were then converted into georeferenced SEGY format using the free computer program SeisPrho [60], distributed by ISMAR- CNR (http://software.bo.ismar.cnr.it/seisprho, accessed on 19 October 2017).

The high-resolution dataset was also integrated with multichannel seismic profiles part of the "Reconnaissance Seismic" database, progressively acquired by Agip S.p.A., on behalf of the Italian State, according to Law, 21 July 1967, n. 613 (https://www.videpi.com,

accessed on 12 October 2017). These data were acquired during the 1980s, off-shore of northern Sicily in the area called "Zona-G" (black lines, Figure 2B). The energy source was an array of airguns with a total capacity of 2000 cubic inches; the receiver was a 2400 m long streamer, with 96 channels spaced 25 m apart. The shot interval was 25 m, allowing a coverage of 4800%. Seismic signals were recorded down to 8 s TWT bsl and processed up to CDP (Common Depth Point) stacking. According to Italian legislation, this dataset is now in the public domain and easily downloaded in PDF format from the ViDEPI project website (https://www.videpi.com). We then converted seismic images from raster PDF to SEGY format, as described above, in order to merge them with all available seismic profiles in a unique georeferenced Kingdom (HIS Markit) project (Figure 2B).

To better constrain the very deep structures, we decided to insert in this work the deep penetration multichannel seismic profile CROP M6A (yellow line in Figure 2B), which was processed up to a time migration down to 17 s TWT bsl. CROP data were collected at the beginning of the 1990s under an agreement between CNR, Agip S.p.A. and Enel (originally National Electricity Authority) as part of a multidisciplinary research program aimed at understanding the main geodynamic processes responsible for the current geological setting in the Italian territory and its surrounding seas [61]. The energy source consisted of 4 tuned arrays of 32 airguns, with a total capacity of 4906 cubic inches; the receiver was a 4500 m long streamer, with 180 channels spaced 25 m apart. The shot interval was 50 m, allowing a coverage of 4500%. The primary positioning system was a GPS Trimble 4000A. CROP data, both raw and processed, are available on request from the CROP Data Centre website (www.crop.cnr.it, accessed on 11 May 2017).

### 2.3. Magnetic Data

The magnetic data were collected during the TIR96 cruise, within the framework of the National Research Council and "Dip. Servizi Tecnici Naz. - Pres. Consiglio" funded project "Lithosphere Formation in mid-oceanic ridges and back-arc basins: Geological Studies in the Equatorial Atlantic and Tyrrhenian Sea", onboard the R/V Gelendzhik [62].

The magnetic anomaly map herein presents the results of reprocessing the ca. 4200 km of lines acquired (green lines, Figure 2C) orthogonally and parallel to the Sicilian margin. Raw data were corrected for spikes and diurnal variations using the reference station of L'Aquila (central Italy). The remaining artifacts in magnetic data were removed by applying the lag and the heading corrections. Finally, we leveled the data using the crossover errors estimation among parallel and oblique lines. Magnetic anomalies were calculated by subtracting the IGRF-12 (International Geomagnetic Reference Field; [63]) model and then reducing the data to the North Pole by phase shifting them using the inclination and declination values as derived from the IGRF.

### 2.4. Seismostratigraphy

In order to produce a reliable structural model, bathymetry was plotted together with the location of the seismic profiles (Figure 2). Using Kingdom's tools, we identified on seismic lines the main faults and the key horizons. Magnetic anomalies allowed us to discriminate where volcanic rocks were present. Based on the seismo-stratigraphic character calibrated by the ODP (Ocean Drilling Program) wells (see details in [25]) and on previous works [23,32,64–67], we identified three main units, represented by (i) Plio-Quaternary sediments, (ii) Continental Basement or (iii) Volcanic rock when associated with magnetic anomalies.

Figure 3 shows in detail the seismo-stratigraphic character of the main units present. The continental Basement is characterized by sporadic and weak internal reflections, usually bounded upward by a strong, high amplitude reflector. The Basement corresponds to the Sicilian-Maghrebian folds-and-thrusts belt, extending on land in northwestern Sicily, [65] with outcrops of the Panormide carbonate platform unit [66]. This unit has typical low magnetic anomaly values, usually around $\pm$ 30 nT, as pointed out in the A-B profile extracted along the seismic profile SS19 (Figure 3).

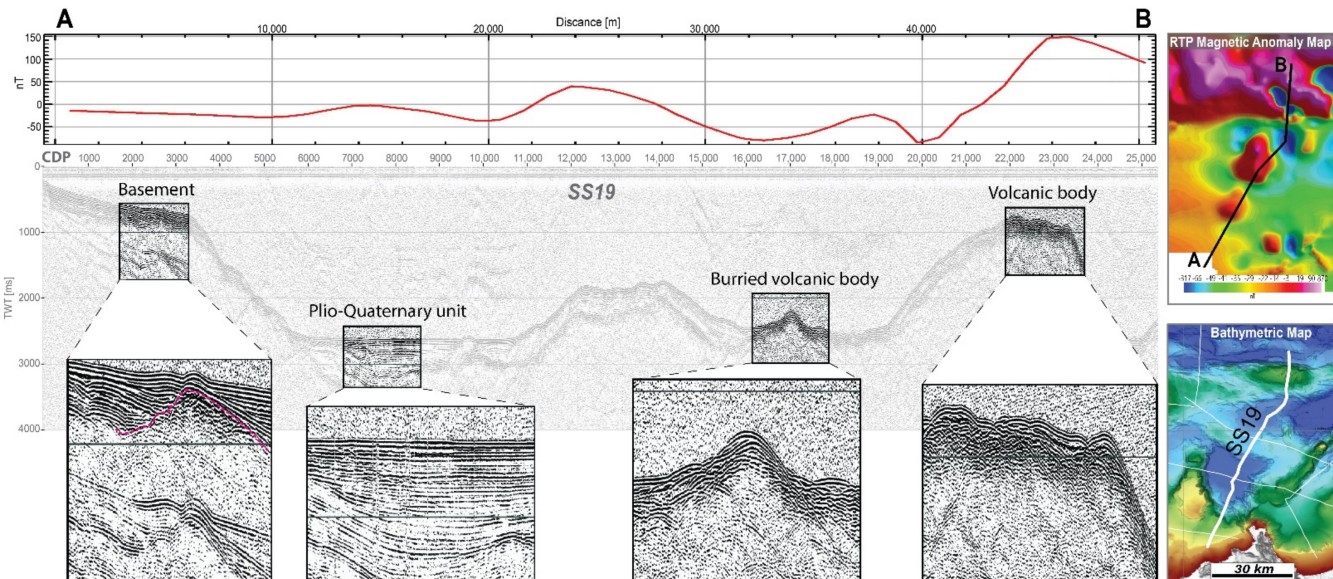

**Figure 3.** Seismo-stratigraphic character of the main units recognized in the study area. Dottom: 30 kJ Sparker profile showing the main units present in this part of the Sicilian margin (bottom). Top: Magnetic profile (A–B in top right side) extracted along the seismic profile navigation (bottom right side). Maps of magnetic anomaly and of bathymetry showing the location of the magnetic profile (A–B) and seismic profile (SS19), respectively (right side).

The Plio-Quaternary unit (Figure 3) corresponds with high-amplitude, laterally continuous and well-stratified reflectors. The upper part of this unit is usually highly reflective with respect to its deeper part. Indeed, the upper part is associated with Pleistocene deposits characterized by higher sand/silt fraction, while the less reflective lower part is associated with Pliocene deposits within which the silty clay fraction is increased (see [68]). This unit is characterized by very low magnetic anomaly values (see A–B profile). We observed that the recent deposits are characterized by the presence of numerous gravitational-related bodies, likely tectonically-induced, sourced from the Sicilian continental slope. The internal reflections of these bodies, interbedded within flat reflectors, are undulated and slightly discontinuous.

Volcanic bodies buried under sediments are difficult to recognize due to a seismo-stratigraphic character similar to undifferentiated basement or deformed sediments, i.e., low internal reflectivity, highly discontinuity and widespread hyperboles of diffraction (Figure 3). We tentatively inferred them by using the magnetic anomalies distribution. High magnetic anomaly values (> ±60 nT) associated with low reflective bodies (see Figure 3), this last tapped by reflectors more or less continue but highly reflective, suggest the presence of buried volcanic bodies.

An example of the possible presence of a volcanic body or volcanic edifice is shown in the last box of Figure 3, where the body is characterized by a chaotic and very high amplitude reflectivity, inside of which no stratifications are recognizable. This reflectivity is very intense in the upper part, while at depth no other reflections are present. This unit is then covered by a very thin layer or a single horizon draping it. The magnetic anomaly computed along the seismic line shows very high values of about 150 nT. Based on these elements, and considering the seismic character of volcanic unit sampled in ODP 655 drill-well [23] and further summarized in [25], at the top of Gortani seamount and along the Palinuro Volcanic Complex [69], we may consider this seismo-stratigraphic phase as diagnostic of volcanic edifices.

## 3. Results

### 3.1. Magnetic Anomalies

Figure 4 shows isobaths of seafloor overlapped on the reduced-to-the-pole magnetic anomalies acquired along Sicilian offshore area. High positive and negative ($> \pm 50$ nT) values are widespread over the seamounts, from Priamo to Prometeo and northward. The Drepano seamount, in literature considered of volcanic origin [70], is characterized by low magnetic anomaly, ranging from 20 to ca. $-40$ nT, except for the southwest side, where detected values are over 90 nT. Moving to the continental slope, the magnetic anomalies are generally low, ranging from $-40$ to 40 nT, and correspond to a green–yellow–red colour scale. High magnetic anomaly positive and negative values are present within basins around Anchise Smt, Ustica Island and Prometeo Smt; a local anomaly is also present offshore San Vito Lo Capo (Figure 4).

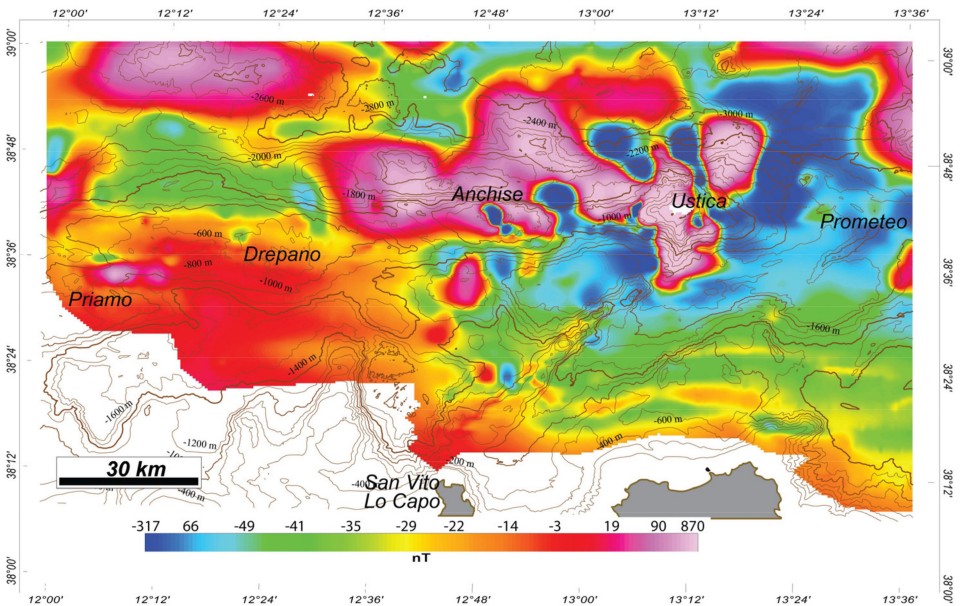

**Figure 4.** Reduced-to-the Pole magnetic anomaly map superimposed on the seafloor isobaths.

### 3.2. Morpho-Bathymetry

Seismic and bathymetric data allowed us to identify a series of basins laterally confined by highs or elongated continental blocks. Highs, blocks and basins have variable orientation: NW–SE, E–W and NE–SW, among which this last is dominating (see dashed lines in Figure 2A). Between 13° and 13.5° E, a 40 km long morphological escarpment (see red arrows in Figure 2A) cuts the seafloor in a NE–SW direction and bounds a canyon that shows a deviation of the pathway in correspondence of a morphological step (see red arrows in Figure 2A). The seafloor of the E–W-trending Drepano-Ustica Ridge is widely incised by WNW–ESE, ENE–WSW up to N–S morphological steps, some of which bound small eye-like basins to the south of the Drepano ridge. Within some basins, for example, in the Medea Basin (MB in Figure 2A), the seafloor shows small undulated ridges (see white arrows in Figure 2A). Finally, several slide and slump scars, not easily observable due to the resolution of the data, are eroding the slopes that bound the basins (white and black arrows in Figure 2A).

### 3.3. Seismic Profiles

Figure 5A shows the sparker profile SS19, acquired across the Erice Basin (Figure 2B). North of Sicily, the continental basement is thinned by NE-dipping normal faults. These faults are responsible for the formation of a very deep basin, namely Erice Basin, filled by the Plio-Quaternary unit. These sediments are folded and faulted by a double verging fault-propagating-fold and by a positive flower structure. Deformation locally reaches the

seafloor (see pink horizon in Figure 5A). The Erice Basin is confined northward by another structural high covered by a thin layer of Plio-Quaternary sediments. Deformed sediments filling the basin are also detected in the obliquely crossed profile G82-88 (Figure 5B). Here, despite the presence of numerous diffractions masking the real geometry of horizons, the sediments are folded and form an anticline that grows at the border of San Vito Lo Capo Ridge. This is in turn affected by NW-dipping normal faults, responsible for the formation of the basin, which here reaches 2100 ms TWT-depth.

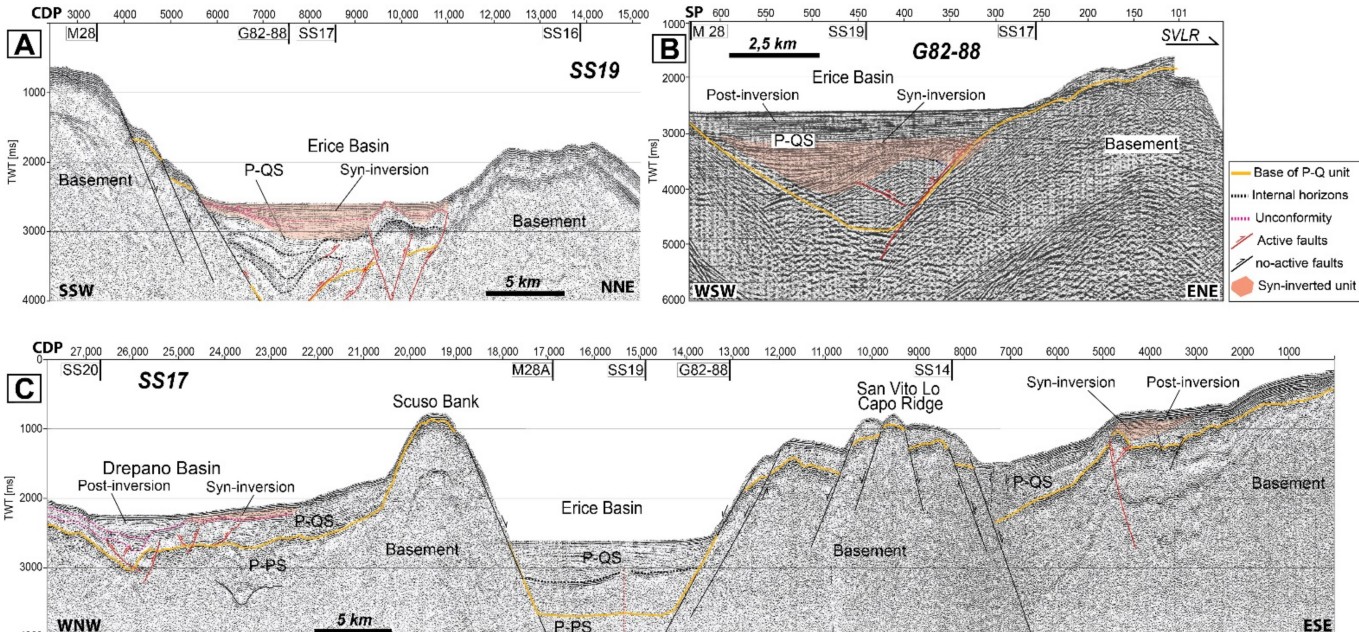

**Figure 5.** Seismic profiles from locations in Figure 2B. (**A**) High-resolution 30-kj sparker profile SS19. P-QS: Plio-Quaternary Sediments. (**B**) Multichannel seismic profile G82-88, acquired by Italian Government. (**C**) High-resolution 30-kj sparker profile SS17. P-PS: Pre-Pliocene Sediments. Syn-inversion sediments [69].

Sparker profile SS17 (Figure 5C) crosses, from continental basement to Drepano Smt, Erice Basin, San Vito Lo Capo Ridge and Scuso Bank (see SVLR and SB in Figure 2A,B). The main features are represented by two structural highs, the SVLR and SB, which are shaped by large E- and W-dipping normal faults, which thinned the basement, allowing the formation of the deep Erice Basin. These faults are also responsible for the formation of minor basins like the Drepano Basin (Figure 5C). These large faults displaced the Plio-Quaternary sediments, which locally thickens, recording a syn-tectonic sedimentation (see CDPs 7000 to 10,000). On the other side, the upper part of the sediments filling Erice Basin are well stratified and undeformed, and they lay in disconfomity above a strong and continuous reflector, marked in Figure 5C with a dashed black line (see CDPs 13,000 to 18,000). Here, deep sediments lack reflectivity or, if present, are parallel, suggesting the sparker profile has been acquired ca. orthogonal to the vergence of structures. In the centre of the basin, a single small sub-vertical discontinuity was detected and is marked with a red dashed line in Figure 5C. Scuso Bank bounds to WNW the Drepano Basin, which is characterized by a thick P-Q deposit of ca. 500 ms that appears affected by compressive deformations confined below seafloor.

Sparker profile SS20 (Figure 6), acquired on the western side of the studied area (Figure 2B), highlights the continental basement covered by a variable thickness of Plio-Quaternary sediments. P-Q sediments are widely folded and faulted, mimicking a series of positive flower structures. Deformation reaches the seafloor and is concentrated in the Drepano Basin. Southward, along the continental slope, the Basement and the overlying P-Q unit are deformed by another positive flower structure. Between these two folded

zones, a large normal fault, apparently south-eastward dipping, has thinned the Basement, allowing the formation of Leda Basin, infilled by about 1280 ms-TWT of P-Q sediments.

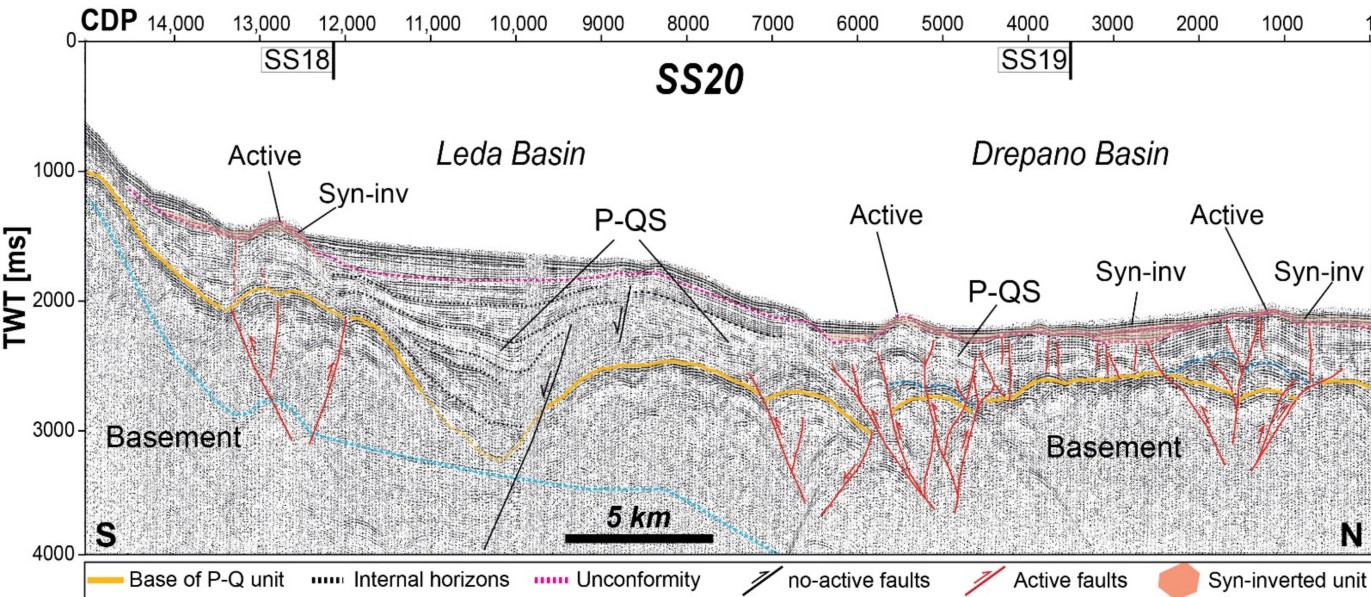

**Figure 6.** High-resolution 30-kj sparker profile SS20, from location in Figure 2A. P-QS: Plio-quaternary Sediments; Syn-inv: Syn-inversion sediments.

South of Ustica, the sparker profile BC34 (see Figure 2B for location) investigates a segment of thinned and intruded continental margin from the Anchise to the NE-trending continental escarpment, crossing the Ustica and the Medea basins, which are separated by the San Vito Lo Capo Ridge (Figure 7A). The deposits at the top of the Anchise Smt are affected by sub-vertical faults, unfortunately not imaged at depth. In the central part of the profile, although San Vito Lo Capo Ridge is less reflective, E- and W-dipping normal faults are detectable. Here the magnetic anomaly has a value of about −70 nT (Figure 4), allowing us to hypothesise the presence of magnetic bodies buried below a layer of PQ sediments. The two basins are affected by compressive deformations by a fold propagation fault, with single or double vergence, as in the Medea Basin. Compressive deformations affect both the Basement and the Plio-Quaternary sediments, but never affect the seafloor, even if recently active in the Medea Basin (see pink unconformity in Figure 7A). Normal faults responsible for the two basins' formation are not detectable, except for the SE side, where several NW-dipping faults thin the basement and cut Plio-Quaternary sediments that thicken against fault plains.

Sparker profile BC33 (Figure 7B) shows the sediments deposited within the San Vito Canyon passing laterally into the Medea Basin (Figure 2A), where they accumulate in contemporary to the activity of NW- and SE-dipping normal faults. These sediments are folded and faulted above the NW-dipping normal fault, which thus become inverted (inset 1); the null point [71] was marked using the geometry of horizons (underlined with black dashed lines). A second minor fold-propagation-fault structure cuts the main fold, which, since it reaches the seafloor, becomes currently active, while the main fold appears to have recently stopped the activity. A large NW-dipping normal fault has shaped the Basement and left a long NE-trending escarpment, currently marked by copious slide and slump events (white and black arrows in Figure 2A). To the rear of this escarpment, further SE-dipping normal faults cut the Basement and the Plio-Quaternary unit, resulting in a Horst and Graben structure. On the other side, San Vito Lo Capo Ridge is cut by two large SE-dipping normal faults.

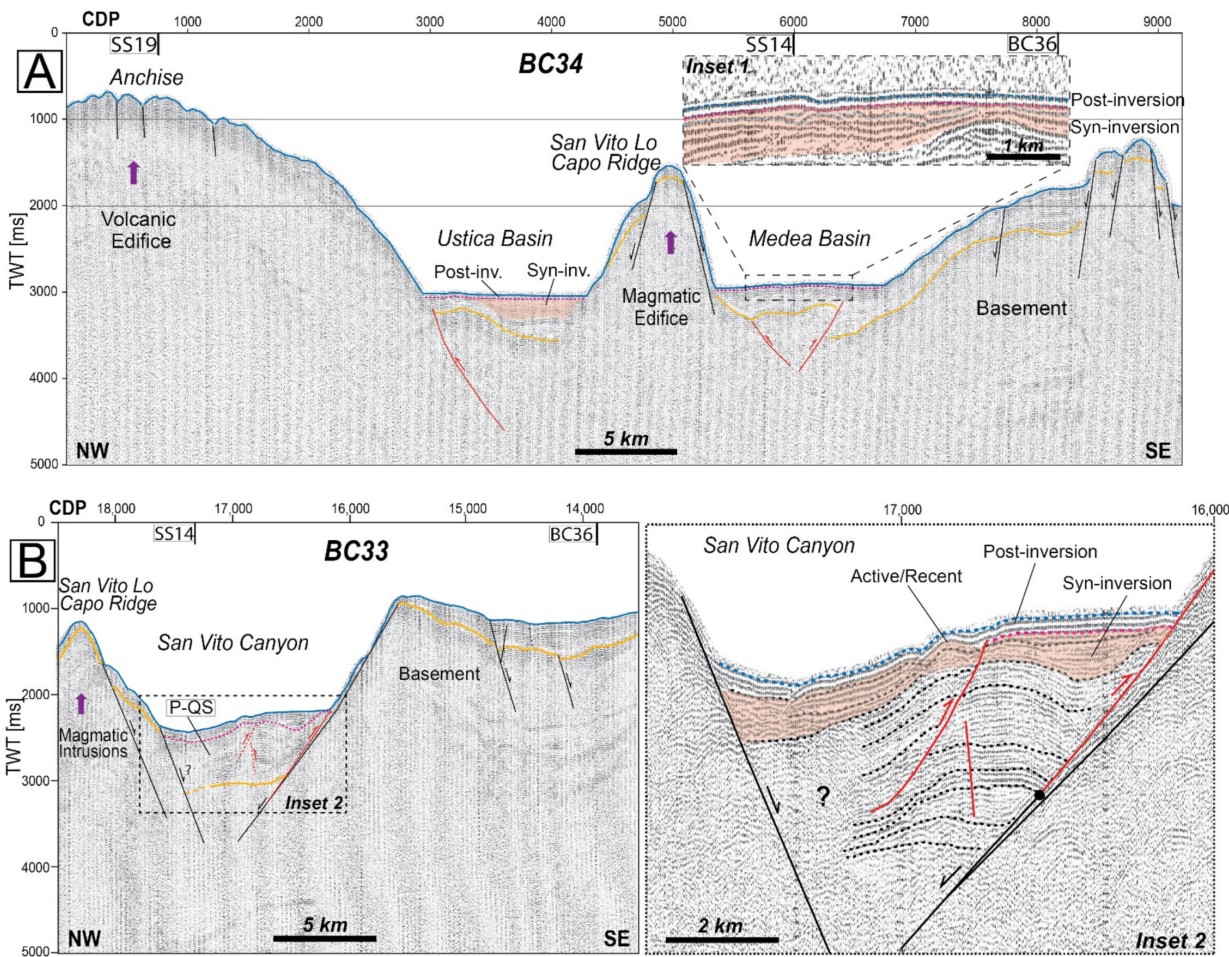

**Figure 7.** Seismic profiles from locations in Figure 2B. (**A**) High-resolution 30-kj sparker profile BC34. (**B**) High-resolution 30-kj sparker profile BC33, crossing San Vito Canyon. Dashed black rectangle outlines section shown on the right side (Inset 2).

### 3.4. Structural and Sedimentary Thickness Map

The structural map shown in Figure 8 synthetizes the interpretation of seismic profiles, bathymetry and magnetic anomalies. The main structural features are represented by a series of NW- and NE-trending normal faults, currently not active (marked with white colour) that bound the several continental highs as well as positive flower-like structures or SE-verging active compressions that are widespread within the sedimentary basins. Main bodies of volcanic ridge are separated by the NW- and the E-W-trending normal faults that locally allowed the formation of eye-like shape basins (white arrows in Figure 8). To the west of Ustica, a series of E–W elongated morphological escarpments, observed on the morpho-bathymetry, are interpreted as fault-controlled scarps belonging to the Arso faults system; see also [72]. Unfortunately, no seismic profile crosses this part of margin to confirm the presence of these faults and their movement, i.e., if normal or strike-slip. The compressive structures, currently active and marked with red colour in Figure 8, are localized inside the sedimentary basins with a mainly NE–SW orientation, except for the deforming sediments filling Erice Basin. Finally, we also plotted the compressive focal mechanisms of the earthquakes recently recorded (see Figure 1B), in which epicentral locations fall within basins hosting compressions and at the border of San Vito Lo Capo Ridge.

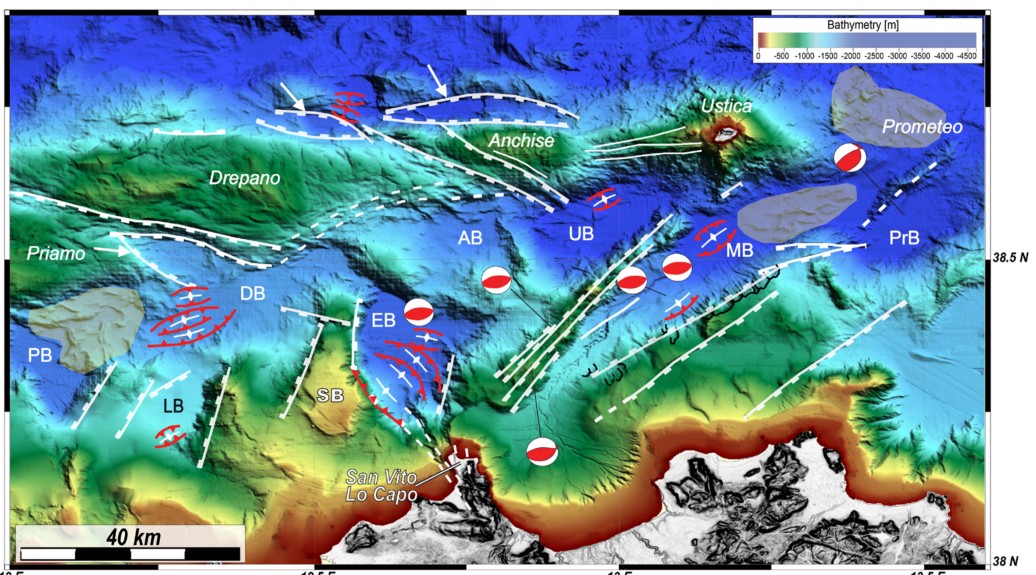

**Figure 8.** Morpho-tectonic map of the northwestern Sicilian margin. Normal faults no longer active are marked with white; inverse faults currently active are marked with red; faults with no well-defined character are marked with dashed white lines; slide scars are marked with black. Transparent masks underline that areas hosting volcanic deposits that have deformed the seafloor. PB: Priamo Basin; LB: Leda Basin; DB: Drepano Basin; AB: Anchise Basin; UB: Ustica Basin; MB: Medea Basin; PrB: Prometeo Basin; EB: Erice Basin; SB: Scuso Basin.

A regional map of the thickness of sediments was created based on the interpretation of all the seismic profiles analyzed in this paper (Figure 9). We digitized the top of the basement/base of the P-Q unit in the seismic profile, using the Kingdom software; then, we calculated the difference between this layer and the seafloor. The thickness, in seconds, of the P-Q unit was converted in meters using an average seismic velocity of 2000 m/s. We plotted the sedimentary thickness values in the software Global Mapper, and in order to ensure uniformity of the data, to create a homogeneous map and to define accordingly the lateral extension of basins, we drew by hand the isopach, based on the morpho-structural basemap (Figure 8). The mean value of the thickness of sediments ranges from 300 to 200 m (orange areas in Figure 9). Low values (<100 m) were detected along the eastern part of the Drepano-Ustica Ridge (Anchise Smt. and Ustica islands), at the top Anchise Basin (AB in Figure 8), and at the top of highs of San Vito Lo Capo Ridge and Scuso Bank. Very high values of sedimentary thickness (>1500 m) occurred north of the Ustica ridge, within Drepano and Erice basins, and to the northeast of San Vito Lo Capo.

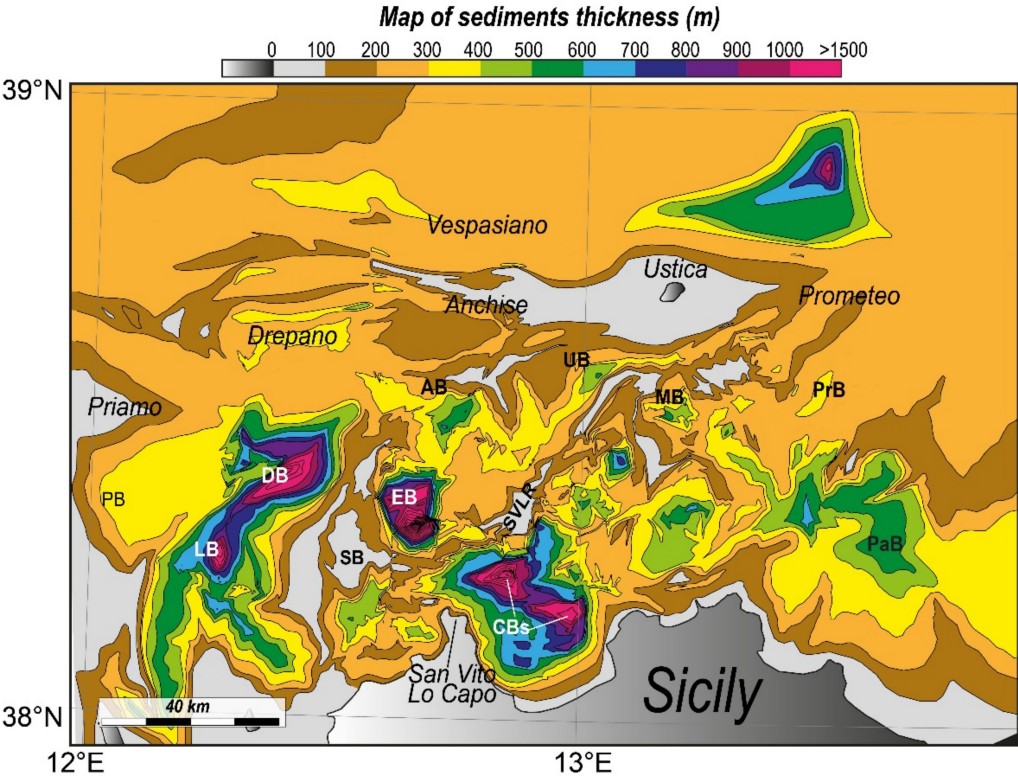

**Figure 9.** Regional map of thickness of sediments along the northwestern Sicilian offshore. PB: Priamo Basin; LB: Leda Basin; DB: Drepano Basin; AB: Anchise Basin; UB: Ustica Basin; MB: Medea Basin; PrB: Prometeo Basin; PaB: Palermo Basin; SVLR: San Vito Lo Capo Ridge; CBs: Castellamare Basins; EB: Erice Basin; SB: Scuso Basin. Contour interval, 100 m.

## 4. Discussion

The northwestern Sicilian margin, corresponding to the southern part of the Tyrrhenian back-arc basin, although very interesting, has not been well studied. Considering its geographical location, i.e., south of Vavilov and west of Marsili basins (Figure 1A), here we linked its history to the evolution of these two oceanic basins. We identified two main tectonic phases that have shaped the northwestern Sicilian margin: (1) the extensional Pliocene phase, dominated by the opening of the back-arc Vavilov sub-basin, and (2) the contractional or transpressional Quaternary phase, dominated by Africa-Eurasia convergence.

### 4.1. Basin Opening in Pliocene Time

During Pliocene, the entire central part of the Tyrrhenian BAB underwent extension, which led to the opening of the Vavilov Basin [25,64 and references therein]. The Vavilov Basin is bounded to the south by several morphological features, amongst which the most relevant is the Drepano-Ustica Ridge (D-UR), indicated by some authors to be associated a wide dextral shear zone [73,74].

The Drepano-Ustica Ridge (D-UR) separates the NW- to NE-trending elongated basins located in the Sicilian margin from the Vavilov basin that opened due to the W–E retreating slab [14,75,76]. D-UR formed during Pliocene, as inferred by the age of rocks dredged at the top of Aceste and Drepano seamounts (from 5.3 to 3.5 Ma; [23]). During the Pliocene, the D-UR probably played a relevant role in the evolution of the southern-central Tyrrhenian basin, as also highlighted recently by [77]. [23] suggested that this ridge of volcanic origin represents the relict of the calcalkaline volcanic arc that formed during the opening of the back-arc Vavilov sub-basin; however, [78] suggests the Ustica Is. as the southern end of an intermediate volcanic arc formed during Pliocene. Calcalkaline rocks have been sampled only at the top of Anchise Smt. [70]. Moreover, due to the morphological shape, and supported by compressive earthquake activity recorded during last decades in this area

(Figure 1B), the D-UR has been interpreted as an internal Miocene thrust named Drepano Thrust Front [79,80], along which Africa-Eurasia convergence was accommodated [81]. Nevertheless, the seismic data so far presented by these authors do not show evidence of thrusts cutting the base of volcanoes (Figure 7A) Instead, the Ridge is affected by several E–W, NW–SE and NE–SW oriented morphologic features (see also [74]), most of which correspond to normal faults (Figures 7A and 8). Based on morphotectonic analysis performed by [74], the NE–SW faults were active during Pliocene, while the NW–SE and E–W faults responsible for the eye-shaped basins opening were active mainly during Pleistocene. Despite the presence of some anticlines detected within sediments covering the ridge [27], D-UR is formed during an extensional regime able to control the deep mantle up-welling along a lithospheric discontinuity, as inferred from the OIB-type magmatism of the rocks dredged along the Aceste, Drepano and Ustica Sms [70,82]. Thus we may consider this long Ridge as controlled by the activity of a very long STEP fault, similar to the Palinuro Volcanic Complex [36,41].

At the time of the opening of the Vavilov basin, the presence of the D-UR paleo-STEP fault is in agreement with sinking and Eastward-retreating slab, which produce a lithosphere tear. This slab geometry may explain the presence of a very large STEP fault in the south-central Tyrrhenian basin and is in agreement with STEP fault models proposed by [38] and by [11]. The paleo-STEP fault we propose merges with the Eolo–Enarete–Sisifo tear fault proposed in [36], of which Ustica Is. is a part.

The D-UR paleo-STEP fault played a fundamental role in the dynamics and kinematics of the surrounding areas, for instance, in the formation of the elongated basins developed to the south. During Pliocene, the lithosphere tear induced by the slab retreating propagated eastward as a scissor-type fault, inducing a drag force that could trigger rotational movements on its southern edge (Figure 10A,B). This model is in agreement with the model proposed by [11], where the fixed limbs are affected by rotations (see Figure 10B). Intense rotations have been also reported southward, in the Sicilian hinterland, derived from paleo-magnetic measurements [83–86]. This hinterland region recorded 63° of total clockwise rotation, 37° of which was attributed to the Pliocene-Quaternary period [87], due to the NW migration of Africa with respect to Eurasia. It appears that the northwest Sicilian offshore underwent high rotation due to both movement NW/NNW of the Africa plate and lithospheric tearing to eastward retreating Ionian slab (Figure 10A,B). This rotation fragmented the continental basement of the Sicilian offshore and favoured the opening of elongated basins bounded by NE-trending normal faults (Figure 10B). Since direct paleo-magnetic measurements are not available in the offshore, we considered in the deformational model (Figure 10), as maximum rotational values experienced during Pliocene, the ones corresponding to the double of rotational values measured in the hinterland area (i.e., 19°), using rotational rates of 13.92°/Ma. Thus, the total clockwise rotation of the offshore area during Pliocene was estimated to be about 38°.

### 4.2. Basin Inversion in Quaternary Time

At the beginning of the Pleistocene, the entire subduction system migrated SE, inducing the opening of the back-arc Marsili basin by the sinking and roll-back of Ionian slab [24]. At this time the D-UR STEP fault was no longer active, except for the Ustica volcano, in which subaerial OIB-type magmatism occurred from 0.79 [88] to 0.132 Ma [89]. This does not exclude the possibility that magmatic activity in the inner and deeper part of the volcano may have started earlier, during Upper Pliocene (3.5 Ma). Accordingly, starting from lower Pleistocene, the offshore of northwestern Sicily was affected only by the NW/NNW push of the African Plate, responsible for the generation a widespread compressional stress field, currently active as confirmed by GPS measurements [29,52]. The compressive stress field, recorded by the deformation of sediments to north of Sicily, is active mainly during Quaternary, as shown by inverted structures in the San Vito Canyon (Figure 7B), while on the rear of the Maghrebian-Appenine chain, between Tunisia and Sicily, the compression has been active intermittently since Pliocene [28,90].

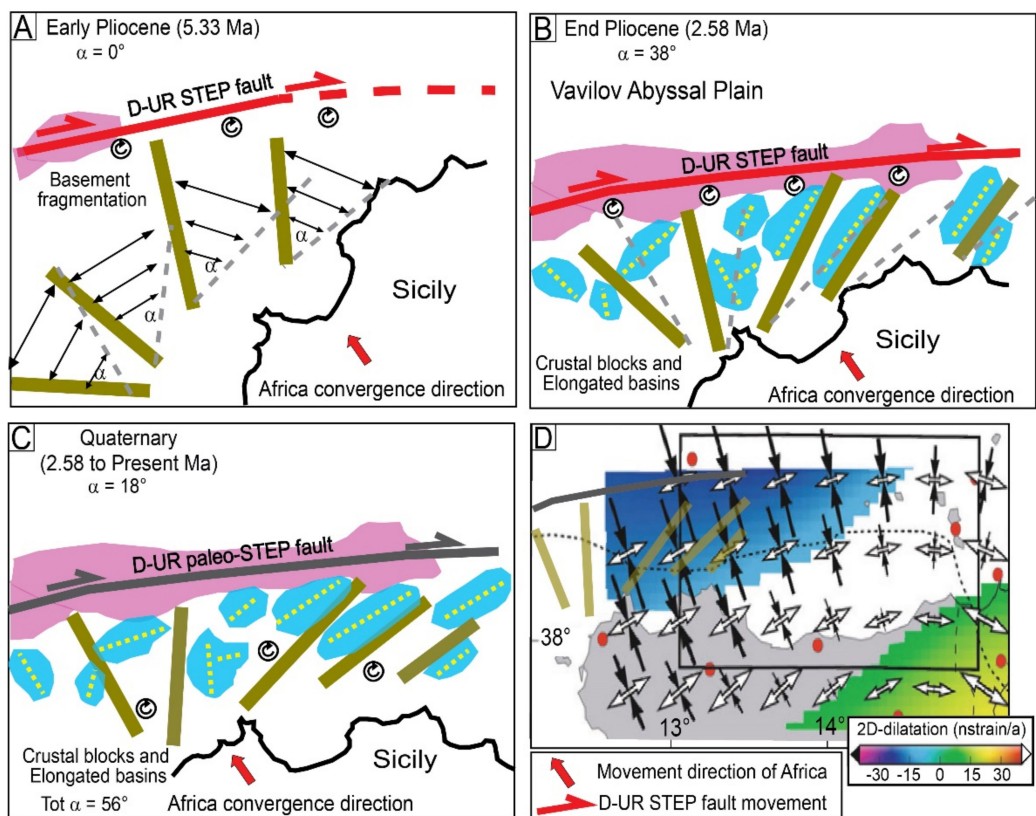

**Figure 10.** Images of the continental blocks rotational model of the area narrowed between D-UR and Sicily. (**A**) Beginning of the Pliocene phase. (**B**) End of the Pliocene phase, which corresponds with the beginning of the Quaternary one. (**C**) End of the Quaternary phase. Red thick arrows mark the movement direction of Africa. Green thick lines correspond to continental blocks. Dashed grey lines in (**A**) mark the position of continental blocks at the present time, (**C**). Blue polygons correspond to the elongated basins, while yellow dashed lines mark the major axes of them. Basin inversion in Quaternary time. (**D**) GPS Strain rate field; black arrows and white arrows mark shortening and extension principal axes, respectively (modified by [29]).

Sediments infilling the elongated basins underwent a tectonic inversion, with the formation of anticlines, fold propagation faults with single (Ustica basin and San Vito Canyon; Figure 7A,B) or double vergence (Erice Basin; Figure 5A), and positive flower-like structures (Drepano and Leda basins; Figure 6). According to [71], these inversion structures affect the entire sedimentary section deposited since Pliocene and are partly covered by syn-inversion and post-inversion Quaternary deposits. Deformation locally reaches the seafloor, suggesting very recent activity, in agreement with instrumental seismicity (Figures 1B and 8) and with strain field derived by GPS measurements (Figure 10D; [29,53]). Moreover, considering that in this part of the margin the paleo-STEP fault drag forces stopped at the end of Pliocene, we may assume that the tectonic inversion started in the lower Pleistocene and continued up to the present (Figures 5A, 6 and 7B). Inversions of sedimentary basins or of inherited normal faults (Figure 7B, inset) to the northwest and to the west of Sicily have been already observed by [27,28,30,32]. Our work suggests that the tectonic inversions of the sedimentary basins are widespread in the entire Sicilian offshore from Ustica (Figure 1) to the Tunisia-Sardinia Escarpment (TSE; Figure 1).

Basin inversion is a topic largely studied, and several factors of control have been defined: (1) orientation and frictional properties of the basin-bounding fault systems [91] and references therein]; (2) presence of fluids that can generate overpressures, reducing the internal frictional coefficient; (3) sediments mechanically weaker than surrounding rocks; (4) sediments deeply weakened by a previous extensional shear strain phase [92]. In our study area we found most of the conditions listed above. The area has been deeply

weakened by a previous extensional shear strain. Deformed sediments within the basins are bounded by normal faults, and inversion structures are all located very close to normal faults (Figure 5A,B, Figures 7B and 8). Between the continental blocks and sediment filling basins, there is a large difference in petro-physical properties: the blocks are part of the Triassic Panormide carbonate platform, while basins are filled by terrigenuos marine sediments enriched with clays and/or pore fluids that further weaken sediments, making them highly deformable. Furthermore, oblique orientation, ranging from 7° to ca. 90° (Figure 10C), between the direction of the compressive stress field, moving NW/NNW (thick red arrow; Figure 10), and the normal faults bounding rigid blocks allow inversion of pre-existing normal faults. Although this angle range is larger than the one proposed in [93], inversion of inherited faults could have been favoured by low friction coefficient of sediments with respect to the continental blocks at the fault plane [90,92].

According to the model proposed herein (Figure 10C), Africa moves, rotating the rigid continental blocks, which in turn compress sediments deforming them in anticlines and positive flower structures. In fact, compressive earthquakes are mainly located along block sides bounding deformed sediments (Figure 8). At the current stage, the compression inverted most to all the sedimentary basins, without the generation of thrusts or mega-thrust. Our study suggests that closure of the back-arc Tyrrhenian basin clearly started in its southwestern part, even if it is far from a subduction initiation, and sheds some light on the infant stage of the basin inversion process and on the compressive structure nucleation, a topic that needs further study [80].

Subduction initiation is a process not clearly observed around the world today. Indeed, we can currently observe mature systems or areas affected by compressive stress (Corringe bank; [94]) not yet associated with thrust front or young subducting slab. Subduction initiation has been hypothesized to be active along the southern Scotia margin [95–97] or imminent in the Atlantic [98]; while in the Tyrrhenian basin, our results do not support an imminent subduction initiation. However, our observations allow us to hypothesize that, when the system is mature, subduction will enucleate at the northern side of the D-UR. This bound is the best candidate for the subduction initiation because it is a zone largely weakened, being a fossil STEP fault along which thickened and rigid crust (intruded continental crust) is in direct contact with thin oceanic crust covered by marine sediments. These are the best conditions for the induced-subduction initiation discussed by several authors [99–103]).

## 5. Conclusions

The analyzed area is at the bound between the back-arc Tyrrhenian basin and the Sicilian Maghrebian chain. This area, from the Pliocene to the present, experienced a long rotational history (estimated to be 56°) due to both the NW/NNW movement of Africa with respect to Eurasia and west to east migration of the STEP fault as represented by the Drepano-Ustica Ridge. This area underwent intense extension and rotation during Pliocene, controlled by the D-UR paleo-STEP fault, which led to the formation of elongated and locally very deep basins bounded by fragmented continental blocks, inverted in compression during Quaternary. Compression is documented by sedimentary basins deformed in positive flower structures, anticline with single or double vergence and inverted fault, controlled by the Africa–Eurasia convergence. The tectonic inversion is widely diffuse, from Ustica Is. to the Tunisia-Sardinia Escarpment. The inversion of the back-arc Tyrrhenian basin clearly starts in its southwestern part, even if far from a subduction initiation.

**Author Contributions:** Conceptualization, M.F.L.; methodology M.F.L., V.F., F.M. and C.P.; data curation, V.F. and F.M.; writing—orginal draft preparation, M.F.L.; writing—review and editing, M.F.L., N.Z., C.P., V.F. and F.M. All authors have read and agreed to the published version of the manuscript.

**Funding:** This research received no external funding.

**Data Availability Statement:** Bathymetric data are available in a publicly accessible database EMODnet Bathymetry Consortium (2018): EMODnet Digital Bathymetry (DTM) https://doi.org/10.12770/18ff0d48-b203-4a65-94a9-5fd8b0ec35f6 (https://www.emodnet-bathymetry.eu/) Magnetic data are not freely accessible. Earthquake data are available in a publicly accessible database (https://www.emsc-csem.org/Earthquake/). Seismic data presented in this study are available on request.

**Acknowledgments:** This work was supported by the Italian Consiglio Nazionale Ricerche. Seismic data interpretation was carried out through IHS Kingdom Suite software, freely available to us in the frame of IHS University Grant Program. "GPS information are based on services provided by the GAGE Facility, opearted by UNAVCO, Inc., with support from the National Science Foundation and the National Aeronautics and Spcae Administration under NFS Cooperative Agreement EAR-1724794". We even thank the CROP database (http://www.crop.cnr.it) for providing the seismic data used in our work, and the Ministero dell'Istruzione dell'Università e della Ricerca (grant PRIN 2017KY5ZX8). We thank Geosciences, an open-access journal by MDPI, and the editors for giving us the opportunity to publish the special issue titled "Tectonics and Morphology of Back-Arc Basin". This is ISMAR-CNR, Bologna, scientific contribution n. 2045.

**Conflicts of Interest:** The authors declare no conflict of interest.

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
