# Peer review of "Inverted Basins by Africa–Eurasia Convergence at the Southern Back-Arc Tyrrhenian Basin"

_geosciences, doi:10.3390/geosciences11030117_

Round 1

Reviewer 1 Report

Dear authors,

I have now read your paper "Inverted Basins by Africa – Eurasia Convergence at the Southern Back-Arc Tyrrhenian Basin". I found the paper to be interesting, relevant and generally well presented. The main correction is that I think I minor review of the writing style is needed, particularly in the discussion which is at times hard to follow. Overall, I recommend minor-moderate corrections prior to acceptance. My comments are in the pdf.

Kind regards

Author Response

We thank the reviewer 1 for positive comments and general appreciation of the paper.

We accurately analyzed comments and corrections that reviewer did directly on the pdf file and performed most of the changes he suggested. Below we detailed answers to the reviewer.

In red comments of the reviewer, in black reply to the comments.

With the abstract I felt that more reference to geological ages would make it more understandable for more readers. I suggest adding these for particular events.

We inserted the ages of the main events as suggested.

Line 13: we decide do not insert “and” because the secondary sentence is directly linked with the first part of the first part of the sentence.

Line 15: we inserted “such”

Line 17: we added “Pliocene” at the extensional phase

Line 22: we added “Pleistocene time” to the contractional phase.

Line 30-31: we did all changes suggested.

Line 35: sentence has been changed in “the slab retrating” instead of “subduction”

Line 38: sentence has been changed in “Contractional deformation has been widely ...” instead of “Compressions has been frequently ...”

Line 49: “described” has been used instead of “mentioned”.

Line 99 and 100: we inserted two more references as suggested by the reviewer.

Line 104-105: “with” has been used in place of “for the correct”.

Figure 1: color bar of the bathymetric-depth has been added.

Line 118: we removed the sentence.

Figure 2: the color bar already was in the figure, anyway we enlarged a little bit to make it more visible.

Line 204: we did changes as suggested by the reviewer.

Line 213: we did not change the sentence; we believe that it is written correctly.

Line 2018: Is this methods or results?

Yes, this is a method. We intend to define the seismo-stratigraphic characteristics of specific domains, in order to have elements or a tool to use during the interpretation of seismic profiles.

Line 243: It is presumably not that they are difficult to recognise", it is more that it is problematic to determine what the highs represent.

Yes, we agree with you. But in this section we are just defining a procedure to recognise / discriminate between volcanic bodies, basement and intruded volcanic rocks, in order to better interpret seismic profiles.

Figure 3: Scale hard to read.

We enlarged the number of the scale.

Line 268: we inserted the values “(> ± 50 nT)” we consider high positive and negative values.

Figure 4: The resolution of this figure is quite low?

We increased the resolution of the image.

Line 290-291: we changed a little bit the sentence in order to combine comments of both reviewers.

Figure 6: we inserted a legend explaining interpretation.

Line 365: we removed “deeply”

Line 371-371: we changed the sentence as suggested by the reviewer.

Figure 8: we removed the “?”, it was an error. We modified the dimension of the focal mechanism rescaling it, because even this was an error. All focal mechanisms have same dimension.

Figure 9: What is the contour interval? And why is a discrete colour pallet used?

This kind of representation enhance the small thickness variations on very-large scale, as for the case of yellow-orange-brown areas. Moreover, we edited manually the thickness map produced with kingdom software in order to remove small irregularities and improve areas with a low data coverage. We did this work comparing the map of thickness produced using kingdom software with bathymetric map, method that has been even described in the text. For this reason, we used this kind of colour scale.

In the caption “Contours interval 100 m” has been added.

Line 436: reviewer suggest to remove the word “relevant”, we are sorry but we believe that the Drepano fault has played a relevant role in the evolution of the back-arc basin. Thus, we decided to keep this word.

We inserted “the” before Pliocene as suggested by the reviewer.

Line 439-440: The grammar in this statement is very confusing

We change the sentence in order to improve it.

Line 474: minor changes as done as suggested by the rev. We also changed the word “intense” with “high”.

Line 478-484: This sentence is awkward to read. I suggest rewording.

We rephrased the sentence in order to clarify the meaning.

Figure 10, line 491: we rephrased the sentence to improve its meaning.

Line 498-501: minor editing has been done as suggested by the reviewer.

Line 512: Are they sediments or sedimentary rocks?

In this case we intend to say “sediments”

Line 517: minor changes suggested by the rev have been done.

Line 518: we changed the sentence as “Deformation locally reaches the seafloor suggesting very recent activity in agreement with instrumental seismicity”.

Line 525: we changed the sentence and removed “or recently”.

Line 528: I don't know what you mean by this?

we deleted the words between commas.

Line 536-537: How big?

We do not have measures or information about petro-physical properties of rocks, we only know lithology at large scale of involved rocks. Anyway we changed the sentence to improve its meaning: “Between the continental blocks and sediments filling basins there is a large difference in petro-physical properties”.

Line 547 – 548: minor editing has been done as suggested by the reviewer.

Line 570: minor editing has been done as suggested by the reviewer.

Reviewer 2 Report

This work by Loreto and co-authors on the structure of inverted basins located in NW of Sicily is well organized, uses a high-quality geophysics data set, and makes an adequate structural interpretation that is undeniably interesting for the scientific community.

I believe that the text and figures can be improved if the suggestions that I indicate in the attached pdf are introduced.

As for the discussion, Figure 10 has to be improved as it does not illustrate very well what is referred to in the text.

The last paragraph of the conclusions is only an assumption that is not fully supported by the data presented in this paper and therefore has to be eliminated.

Author Response

We thank the reviewer 2 for positive comments and general appreciation of the paper.

We accurately analyzed comments and corrections that reviewer suggested on the pdf file and performed all changes he suggested. Below we detailed answers to the reviewer.

In red comments of the reviewer, in black our reply to the comments.

All minors editing suggested by reviewer have been done.

Abstract: all changes suggested by the reviewer has been done. As for example, “contractional” has been substitute to “compressional”.

Two key words have been changed has suggested by the reviewer.

Line 29-39: all changes suggested by the reviewer have been done.

Line 65-66: compression has been changed with contraction.

Line 95: sentence has been changed in order to clarify its meaning.

Figure 1B: scale bar has been added.

Figure 2B and C: scale bars have been added. In captions: arrows have been described.

Line 239-240: olistoliths? related to gravitational collapse?

Sentence has been changed: “presence of numerous gravitational-related bodies, likely tectonically-induced, sourced …”.

Line 247: we changed the sentence to clarify its meaning and reply to the reviewer.

Figure 3: we added the scale.

Figure 4. we added the scale.

Line 293: we added description of arrows in the caption of figure 2.

Line 300: Erice graben?

In our case it is Erice Basin. Thus, we prefer to keep basin instead of graben.

Line 308: SVLR : San Vito Lo Capo Ridge , we prefer to keep ridge instead of horst.

Line 315: marking an angular unconformity?

We changed the sentence and described the unconformity that in this case is a disconformity.

Figure 7A inset and B inset: we added the scales.

Figure 8: can you indicate the slide and slump scars

We mapped the slump scars, even if it is little hard to see them at this scale, see black lines

hard to distinguish in the map...

We improved the inverse structures thickening the lines in order to make them more visible.

Caption: we added labels description.

Figure 9: we added the scale. Caption: acronyms description has been added.

Line 417-318: the sentence has been changed to improve its meaning as requested by the reviewer.

Line 431: “dextral” word has been added.

Line 465-466: Dextral motion of D-UR favors closing of NE-SW trending basins and opening of NW-SE basins...

We preferred to keep this part, because we are describing the Pliocene extensional phase, while the contractional phase that led to the closure of the basin happened during Pleistocene is described in the successive paragraph.

Figure 10: we changed the figure make it more readable. We included all suggestions of the reviewer. We added a legend.

Figure 10B: we removed the inset and left the reference in the text to Govers & Wortell 2005.

Figure 10C: we enlarged the inset.  

Caption: we edited the caption to improve it and describe correctly the figure.

Line 523: We write “inherited normal faults” instead of “extensional inherited fault”.

Line 542: we changed accordingly the figure 10.

Line 544: we changed “inherited” with “pre-existing”.

Line 560-562: we changed the sentence in order to clarify its meaning as requested by the reviewer.

Line 570: we changed “suffered” with “experienced”.

Line 574: we changed “paleo-Drepano STEP fault” with “paleo-D-UR STEP fault”.

Line 582-585: we removed the sentence as suggested by the reviewer.
